# Lifelong Learning in StyleGAN through Latent Subspaces

**Adarsh Kappiyath**                                        *kadarsh22@gmail.com*
*TCS Research*
*Delhi, India*

**Anmol Garg**                                        *aganmol1998@gmail.com*
*Indian Institute of Science (IISc), Bengaluru, India*
*TCS Research, Delhi, India*

**Ramya Hebbalaguppe**                                        *ramya.hebbalaguppe@tcs.com*
*TCS Research*
*Delhi, India*

**Dr. Prathosh A. P**                                        *prathosh@iisc.ac.in*
*Indian Institute of Science (IISc),*
*Bengaluru, India*

**Reviewed on OpenReview:** *https://openreview.net/forum?id=I4IAwVOZrM*

## Abstract

StyleGAN is one of the most versatile generative models that have emerged in recent times. However, when it is trained continually on a stream of data (potentially previously unseen distributions), it tends to forget the distribution it has learned, as is the case with any other generative model, due to catastrophic forgetting. Recent studies have shown that the latent space of StyleGAN is very versatile, as data from a variety of distributions can be inverted onto it. In this paper, we propose StyleCL, a method that leverages this property to enable lifelong learning in StyleGAN without forgetting. Specifically, given a StyleGAN trained on a certain task (dataset), we propose to learn a latent subspace characterized by a set of dictionary vectors in its latent space, one for each novel, unseen task (or dataset). We also learn a relatively small set of parameters (feature adaptors) in the weight space to complement the dictionary learning in the latent space. Furthermore, we introduce a method that utilizes the similarity between tasks to effectively reuse the feature adaptor parameters from the previous tasks, aiding in the learning process for the current task at hand. Our approach guarantees that the parameters from previous tasks are reused only if they contribute to a beneficial forward transfer of knowledge. Remarkably, StyleCL avoids catastrophic forgetting because the set of dictionary and the feature adaptor parameters are unique for each task. We demonstrate that our method, StyleCL, achieves better generation quality on multiple datasets with significantly fewer additional parameters per task compared to previous methods. This is a consequence of learning task-specific dictionaries in the latent space, which has a much lower dimensionality compared to the weight space. Code for this work is available at link

## 1 Introduction

Continual learning (CL) is a fundamental machine learning paradigm that focuses on the model's ability to learn and adapt to new tasks or evolving data streams over time while ensuring that previously acquired knowledge remains intact. Extensive research has explored continual learning within the context of discriminative models De Lange et al. (2022), but relatively less attention has been devoted to the application of this paradigm in the realm of generative models. However, recent progress in the field of generative modelling has brought

them to the forefront of application domains. Specifically, models such as Generative Adversarial Networks (GANs) Goodfellow et al. (2014) and denoising diffusion models Ho et al. (2020) have found their utility in a wide variety of tasks such as semantic editing Ling et al. (2021), image in-painting, Yu et al. (2018) etc. Thus, it is imperative to consider the problem of continual learning in the context of generative models Lesort et al. (2019).

It is essential to note that Diffusion models offer remarkable generation quality enhancements, albeit with a trade-off of increased inference time. On the other hand, GANs excel in efficiency, requiring only a single forward pass for generation. Furthermore, the introduction of GigaGAN, Kang et al. (2023) and StyleGAN-T Sauer et al. (2023) have illustrated their ability to provide competitive generation quality while maintaining faster inference speeds in text-conditioned image generation tasks. Given these advantages, we turn our attention to continual learning in GANs for unconditional generation. Also, recently many recent state-of-the-art GANs for various tasks, as seen in works like Kang et al. (2023); Sauer et al. (2023); Fu et al. (2022), employ StyleGAN- based architectures. This inspires our investigation into StyleGAN-based architectures for continual learning.

We hypothesize, that StyleGAN is suited for generative continual learning because of its versatility, in that, a large variety of datasets can be inverted onto its extended latent space ($\mathcal{W}^+$) as observed in Abdal et al. (2019). Motivated by these observations, we investigate whether the latent space of StyleGAN can be exploited to generate data from a stream of datasets without forgetting. Towards that end, we propose a method titled StyleCL to learn a per-task, style-wise dictionary of vectors that define a subspace in the latent space of StyleGAN. In addition to latent dictionary learning, we also learn a set of parameters in the weight space, to accommodate a richer knowledge in tandem with the learned latent subspace.

Knowledge transfer, a cornerstone of continual learning, assumes a central role in StyleCL. StyleCL utilizes the latent space to identify the most similar task unlike GAN Memory Cong et al. (2020) and CAM-GAN Varshney et al. (2021) where the most similar task is characterized using the most recent task or the task with high Fisher information respectively. We also determine the nature of forward knowledge transfer (positive or negative) by measuring the cosine similarity of dictionary vectors to its projection onto the latent subspace of the most similar task which is then used to prevent negative forward transfer.

The following is a summary of our contributions:

- **Latent subspace learning for StyleGAN**: We propose a latent subspace learning approach that enables learning without forgetting for StyleGAN.

- **Improved generation quality**: By harnessing the versatility of the latent space of StyleGAN, our method outperforms contemporary approaches like CAM-GAN and GAN Memory in terms of generation quality, all while employing fewer parameters (28.95% reduction) and FLOPs (11.6% reduction).

- **Prevention of negative forward transfer**: We further propose a simple way to identify the most similar previous task and also characterize the nature of forward transfer between any two tasks to prevent negative forward transfer. To the best of our knowledge, StyleCL stands out as the sole method capable of averting negative forward transfer.

## 2  Related Work

**Generative Continual Learning**: Continual Learning methods are broadly categorized into three categories: replay-based, regularization-based and parameter isolation-based methods. These categorizations are defined for a discriminative continual setting but they can be applied to generative continual learning as well.

Chenshen et al. (2018) introduces MerGAN, a replay-based GAN that combines generated samples from previous tasks with new task data to form an extended training dataset. They also introduce a replay-alignment loss to ensure consistent generation for previous tasks as the number of tasks increases. Zhai. *et al.*, presents Lifelong GAN for continual image-conditioned image generation, employing knowledge distillation and auxiliary data generation by creating patch montages from training batches to mitigate catastrophic

forgetting. However, replay-based approaches face scalability issues due to cumulative inaccuracies when a single generator is incrementally updated.

Parameter isolation techniques like PiggybackGAN Zhai et al. (2020) freeze old task parameters and introduce smaller new parameters for learning without forgetting. GAN memory Cong et al. (2020) employs normalization parameters to adapt the generator's weights to incoming data streams. CAM-GAN Varshney et al. (2021) introduces adaptation modules via group-wise convolutions at the output of each convolution layer in the base network. Similar to CAM-GAN Verma et al. (2021), proposes efficient feature transformation enabling continual learning with minimal parameter expansion for both generative and discriminative tasks. Recently, Seo et al. (2023) proposed LFS-GAN for continual learning in a few-shot setting. This is achieved by learning an efficient task-specific modulator on top of a pre-trained StyleGAN and utilizing these task-specific modulators for generating from each of the tasks during inference. Contrary to all these methods, StyleCL takes a different approach by learning a latent subspace alongside shared weight space parameters, facilitating continual learning rather than solely relying on efficient feature transformations.

Even though few regularisation-based approaches like Liang et al. (2018) and Seff et al. (2017) use regularisations to enable continual learning, their generation quality still degrades over time and thus parameter isolation methods appear to be a better choice and have been receiving more attention.

**Knowledge transfer in continual learning**: Knowledge transfer is a crucial aspect of continual learning, predicated on the notion that similar tasks inherently possess shared knowledge that can be effectively transferred between them. However, previous approaches, like MerGAN Chenshen et al. (2018), Lifelong GAN Zhai et al. (2019), and Piggyback GAN Zhai et al. (2020), often lack explicit mechanisms to facilitate this positive knowledge transfer. While GAN Memory Cong et al. (2020) demonstrates promise in enabling knowledge transfer, it relies on the assumption that the most recent task is invariably the most similar, a notion that does not consistently hold. In contrast, CAM-GAN Varshney et al. (2021) quantifies task similarity by approximating the Fischer information matrix (FIM) and posits that initializing the current task with parameters from the most similar task would consistently yields positive forward transfer which may not always hold true. StyleCL distinguishes itself by characterizing both the most similar task and the nature of forward transfer using the latent space, thus effectively capturing the state of the generator while identifying the most similar task and elucidating the nature of the forward knowledge transfer.

# 3   Proposed Method: StyleCL

## 3.1   Problem Setting

We introduce a model for learning from a sequence of datasets, where each dataset is identified by a unique task ID and is presented sequentially for training. Let $\{\mathcal{X}^t\}_{t=1}^T$ denote these datasets, with $\mathcal{X}^t$ comprising $N$ instances $\{\mathbf{x}_i^t\}_{i=1}^N$, each sampled from the task-specific distribution $p_t$. Our challenge is to obtain a Generative Adversarial Network (GAN) that generates samples from the current task distribution without forgetting the distributions from previous tasks.

Our method starts by training a GAN as in (Karras et al., 2020a) on the first (or base) dataset (task), denoted by $\mathcal{G}^1$. The parameters of $\mathcal{G}^1$ are denoted by $\phi^1$ and are shared by all the subsequent tasks. For each subsequent dataset $\mathcal{X}^t (t > 1)$, we learn a latent subspace characterized by $\mathbf{U}^t$ and $\mathbf{b}^t$ on the latent space of $\mathcal{G}^1$ as detailed in Sec. 3.2. Next, we determine the most similar previous task to the current task, designated as task $k$ by leveraging $\mathbf{b}^t$. We further elaborate this process in Sec. 3.3.1. Upon identifying the task $k$, we acquire its corresponding generator, denoted by $\mathcal{G}^k$. We then evaluate whether the parameters from task $k$ are advantageous for learning the present task $t$ as elaborated in Sec. 3.3.2. When the parameters from task $k$ are beneficial, we learn subspace $\mathbf{U}^t$ and feature adaptors $\phi^t$ within the latent and weight space of $\mathcal{G}^k$, otherwise, we shift this training to the latent and weight spaces of the base generator $\mathcal{G}^1$, as described in Sec. 3.4. Fig. 1 presents an overview of our method, which we name 'StyleCL'.

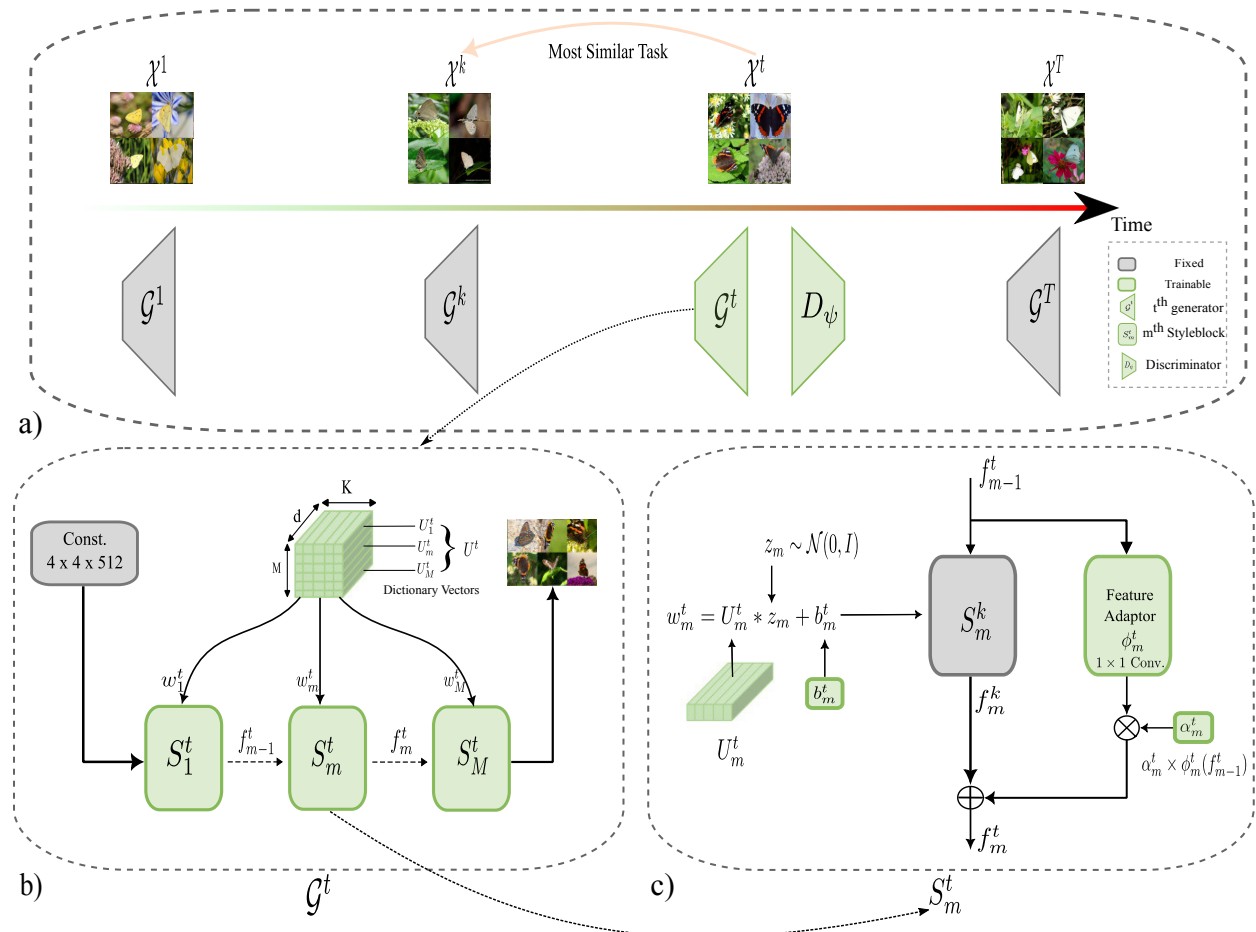

Figure 1: **Overview of StyleCL**: Fig. (a) illustrates the generative continual learning problem, where a continuous stream of data arrives, and the objective is to generate samples from both current and past data without forgetting. At time $t$, the framework begins by learning the parameters $\mathbf{U}^t$ and $\mathbf{b}^t$ in the latent space of the generator $\mathcal{G}^1$ using the dataset $\mathcal{X}^t$, as shown in Fig. (b). Task similarity is then evaluated to identify the most similar previous task $k$, along with its corresponding generator $\mathcal{G}^k$, and the degree of forward transfer is computed. As demonstrated in Fig. (c), the feature adaptor block $\phi_m^t$, along with the parameters of $S_m^k$, is used to obtain the activation map $f_m^t$. If negative forward transfer occurs, $k$ is set to 1, meaning only the style blocks from $\mathcal{G}^1$ ($S_m^1$) are used to generate $f_m^t$.

## 3.2 Latent Dictionary Learning

We employ StyleGAN2 Karras et al. (2020a) architecture for the generators $\mathcal{G}$ that contains $M$ style blocks and for simplicity of discussion, we assume each of these style blocks comprises just 1 layer. The first stage of our method is to learn a set of dictionaries on the extended latent space $(\mathcal{W}^+$ space) of the StyleGAN. Given a dataset $\mathcal{X}^t$, a dictionary $\mathbf{U}_m^t = \{\mathbf{u}_{m1}^t, \dots, \mathbf{u}_{mi}^t, \dots, \mathbf{u}_{mK}^t\}, \mathbf{u}_{mi}^t \in \mathbb{R}^d$, containing $K$ vectors are learned for each of the $m = 1, 2, .., M$ style blocks of the generator along with a bias vector $\mathbf{b}_m^t$. Note that the elements of $\mathbf{U}_m^t$ define a latent subspace within the $(\mathcal{W}^+)$ space of StyleGAN while the learned bias vector $\mathbf{b}_m^t$ serves to translate this subspace. Consequently, $\mathbf{b}_m^t$ characterizes the position of the latent subspace with respect to the origin.

During the training, the parameters of $\mathbf{U}_m^t$ are initialized randomly. First, a batch of vectors is stochastically sampled from each dictionary $\mathbf{U}_m^t$ as follows:

$$\mathbf{w}_m^t = z_{m1}\ \mathbf{u}_{m1}^t + z_{m2}\ \mathbf{u}_{m2}^t \dots + z_{mK}\ \mathbf{u}_{mK}^t + \mathbf{b}_m^t \tag{1}$$

where $\mathbf{z}_m = [z_{m1}, \ldots, z_{mK}] \sim \mathcal{N}(0, \mathbf{I})$. Further, each $\mathbf{w}_m^t$ corresponding to every style block is concatenated to form $\mathbf{w}^t$ as:

$$\mathbf{w}^t = [\mathbf{w}_1^t, \ldots, \mathbf{w}_M^t], \mathbf{w}^t \in \mathcal{W}^+ \tag{2}$$

Finally, $\mathbf{w}^t$ is passed as the input to the corresponding generator to generate images from $p_t(\mathbf{x}^t)$.

### 3.3 Forward Transfer

#### 3.3.1 Choosing the most similar previous task

This section outlines a method for identifying the task $k$ that is most similar to the current task $t$. Determining the closest prior task enables the transfer of knowledge from that task to the current task, a concept parallel to forward transfer in continual learning literature Chen & Liu (2018). In order to find the task that is most similar to an incoming task, it is necessary to characterize both the previous as well as the current task in the latent space. We characterize the current task $t$ by learning $\mathbf{U}_m^t$ and $\mathbf{b}^t$ using the base generator $\mathcal{G}^1$. It is important to mention that $\mathbf{U}_m$ and $\mathbf{b}_m$ are already learned for the previous tasks from 2 to $t-1$.

For a given task $t$, we define the most similar task $k$ as the task whose latent subspace is closest to the subspace of task $t$. This assumption is based on the observation that the latent vectors of similar tasks tend to be close in proximity, while those of dissimilar tasks are farther apart, as illustrated in Fig. 5. The bias vector $\mathbf{b}_m^t$ represents the extent to which the latent subspace $\mathbf{U}_m^t$ is shifted, and we can use the Euclidean distance between bias vectors to quantify the similarity between subspaces across tasks. To motivate the comparison of bias vectors as a measure of subspace similarity, consider a simplified scenario where the dictionary consists of a single vector along with a bias term. In this case, the latent vectors lie within an affine subspace, represented as $b + \alpha u_1$, where $b$ is the bias and $u_1$ is the dictionary vector. Similarly, we could define a latent space for another dataset as $b' + \alpha' u_1'$. To measure this similarity, we begin by computing the Euclidean distance between the bias vectors $b$ and $b'$. This distance provides a measure of how closely the subspaces are aligned in the feature space. Furthermore, to assess the alignment between the subspaces more comprehensively, we define a similarity metric $sim(t, k)$ that captures the orientation and overlap of the subspaces defined by the corresponding dictionaries.

The most similar task $k$ is then identified as the task whose bias vector $\mathbf{b}^r$ is closest to the bias vector $\mathbf{b}^t$ of task $t$. Formally, this is expressed as:

$$k = \underset{r:r \in \{2, \ldots, t-1\}}{\operatorname{argmin}} \|\mathbf{b}^t - \mathbf{b}^r\|_2 \tag{3}$$

Intuitively, equation equation 3 quantifies the discrepancy in how the subspaces of task $t$ are translated relative to the subspaces of previous tasks. This choice is grounded in the assumption that subspaces of similar tasks should be closer in the latent space, as evidenced by the empirical results shown in Fig. 5.

#### 3.3.2 Preventing negative forward transfer

Choosing the most similar task facilitates selecting a task with a similar set of features as that of the current task, although this may result in negative forward transfer. This situation arises because two subspaces $\mathbf{U}_m^t$ and $\mathbf{U}_m^k$ might be orthogonal to each other despite being translated by the same bias vector. To alleviate this problem, we estimate the nature of forward transfer (positive or negative) by computing the cosine similarity of dictionary vectors of the current task to its projection onto the latent subspace of the most similar task $k$. To compute the task similarity, both set of latent vectors $\mathbf{U}^t$ and $\mathbf{V}^k$ should be in the same latent space. We learn $\mathbf{U}^t$ in the latent space of $\mathcal{G}^k$ to facilitate this before computing the task-similarity. Let $\mathbf{V}_m^k = \{\mathbf{v}_{m1}^k, \ldots, \mathbf{v}_{mj}^k, \ldots, \mathbf{v}_{mK}^k\}, \mathbf{v}_{mj}^k \in \mathbb{R}^d$ correspond to the orthonormal vectors obtained using the Gram-Schmidt orthogonalization procedure on the dictionary vectors $\mathbf{U}^k$. The projection of $\mathbf{u}_{mi}^t$ onto the latent subspace characterized by the orthonormal vectors $\mathbf{V}_m^k$ for a style block $m$ is defined as:

$$\mathbf{p}_{mi}^{tk} = \sum_{j=1}^{K} \langle \mathbf{u}_{mi}^t \cdot \mathbf{v}_{mj}^k \rangle \mathbf{v}_{mj}^k \qquad (4)$$

Subsequently, the nature of forward transfer is estimated as follows :

$$
\begin{aligned}
sim(t,k) &= \frac{1}{M} \sum_{m=1}^{M} \sum_{i=1}^{K} \cos\left(\mathbf{u}_{mi}^t, \mathbf{p}_{mi}^{tk}\right) \\
&= \frac{1}{M} \sum_{m=1}^{M} \sum_{i=1}^{K} \frac{\mathbf{u}_{mi}^t \cdot \mathbf{p}_{mi}^{tk}}{\left\|\mathbf{u}_{mi}^t\right\| \left\|\mathbf{p}_{mi}^{tk}\right\|}
\end{aligned}
\qquad (5)
$$

The value of $sim(t,k)$ lies in the range $[-1,1]$, indicating the degree of forward transfer. Specifically, $sim(t,k) > 0$ signifies potential positive forward transfer. We do not reuse the parameters from the most similar task when $sim(t,k) \leq 0$ to prevent negative forward transfer. Intuitively, $sim(t,k)$ quantifies the similarity or alignment between the latent subspaces of tasks. $sim(t,k) \leq 0$ denotes orthogonality or higher degree of misalignment between the latent subspaces of the current task $t$ and the most similar task $k$, where feature reuse could be detrimental to the performance. Conversely, higher values of $sim(t,k)$ suggest a strong alignment of these subspaces, which may lead to more effective positive knowledge transfer.

### 3.4 Feature Adaptors in the Weight Space

We observed empirically that learning $\mathbf{U}_m^t$ alone may not be fully sufficient to model $p_t(\mathbf{x}^t)$. Therefore, we also introduce additional feature adaptor blocks on the weight space of the generator to obtain $\mathcal{G}^t$. Since the latent subspace would have already captured some characteristics of the datasets, the number of feature adaptor parameters to be learned would be lesser (Tab. 2). Theoretically, the necessity of feature adaptors arises from the limitation of dictionary learning alone in generating all possible task subspaces. Specifically, dictionary learning may not be sufficient to capture the full range of task-specific features, as the base generator may lack the required features that can be modulated to represent the target features of semantically distinct tasks. By introducing feature adaptors, we extend the capacity of the base generator to learn additional features that are not present or cannot be obtained through modulation of the dictionary features alone.

Let $S_m^t$ denote the $m^{th}$ style block within the generator $\mathcal{G}^t$ for task $t$. We present an approach to compute the activation map $f_m^t$, which essentially corresponds to the output of the style block $S_m^t$. Following Sec. 3.3, we obtain the most similar task $k$ and assess the potential negative forward transfer when considering the utilization of parameters from task $k$ to learn the parameters of the current task. Let $\mathcal{G}^k$ represent the generator corresponding to task $k$. The computation of activation map $f_m^t$ varies depending on whether the task $k$ leads to positive forward transfer or not. Let us consider the scenario in which task $k$ results in a negative forward transfer. In such instances, we refrain from using the parameters of task $k$, and the activation map $f_m^t$ corresponding to task $t$ is obtained using only styleblocks of $\mathcal{G}^1$ as in equation 6. Alternatively, in case of negative transfer, one could resort to identifying the next most similar task, estimate its nature of forward transfer and use its parameters for facilitating forward transfer subject to computational and time constraints.

$$f_m^t = S_m^1(f_{m-1}^t) + \alpha_m^t \times \phi_m^t(f_{m-1}^t) \qquad (6)$$

Here, $\phi_m^t$ corresponds to the trainable feature adaptor which is a $1 \times 1$ Convolution layer. Both $\phi_m^t$ and scaling coefficient $\alpha_m^t$ are learnable and jointly learned with the latent dictionary.

When task $k$ results in a positive forward transfer, we utilize the parameters of task $k$ to determine the activation map $f_m^t$ through the following process :

$$f_m^t = S_m^k(f_m^{t-1}) + \alpha_m^t \times \phi_m^t(f_{m-1}^t) \qquad (7)$$

where

$$S_m^k(f_m^{t-1}) = f_m^1 + \underbrace{\cdots}_{\substack{\text{activations from} \\ \text{similar tasks of } k \\ \text{(if any)}}} + \alpha_m^k \cdot \phi_m^k(f_{m-1}^t)$$

In this scenario, the feature adaptor block $\phi_m^t$ is designed to acquire additional information that is not present in $\mathcal{G}^k$. By adopting this approach to obtain feature activation maps, we ensure that the relevant parameters are reused for the current task, while any irrelevant parameters are discarded. Additionally, given the recursive nature of the computation, it ensures that the parameters of the tasks similar to task $k$ are also employed in obtaining the feature activation maps for the current task $t$.

We follow the training paradigm in Karras et al. (2020a), which includes adversarial loss $\mathcal{L}_1$. We also utilize the Perceptual Path Regularizer (PPL) Karras et al. (2020b) and $\mathcal{R}1$ regularization Mescheder et al. (2018) as in Karras et al. (2020b) to ensure smoothness and facilitate better convergence.

$$\begin{aligned} \mathcal{L}_1(D_\psi, \mathcal{G}) &= \mathbb{E}_{x \sim p_t(\mathbf{x}^t)}[\log D_\psi(x)] \\ &+ \mathbb{E}_{\mathbf{z}_1, \ldots, \mathbf{z}_M \sim \mathcal{N}(0, \mathbf{I})}[1 - \log D_\psi(\mathcal{G}(\mathbf{w}^t))] \end{aligned} \tag{8}$$

We provide a comprehensive overview of the StyleCL training process in Algorithm 1.

---

**Algorithm 1** StyleCL: Training Procedure

---

    **Input:** $\{\mathcal{X}^t\}_{t=1}^T$: Stream of $T$ datasets.
    **Output:** Learned parameters: $\{\mathbf{U}^t, \mathbf{b}^t\}_{t=2}^T$, generator $\mathcal{G}^1$, and adaptors $\{\phi^t\}_{t=2}^T$
1: Train StyleGAN2 with dataset $\mathcal{X}^1$ to obtain $\mathcal{G}^1$
2: **for** $t = 2$ **to** $T$ **do**
3:     Initialize discriminator $\psi$, $\mathbf{U}^t$, $\mathbf{b}^t$
4:     Optimize $\mathbf{U}^t$ and $\mathbf{b}^t$ using $\mathcal{L}_1(D_\psi, \mathcal{G}^1)$ as in equation 8
5:     Determine most similar task $k$ using equation 3 to get $\mathcal{G}^k$
6:     **for each** training iteration **do**
7:         Sample latent vector $\mathbf{w}^t$ using equation 1 and equation 2
8:         Optimize $\mathbf{U}^t$ and $\mathbf{b}^t$ using $\mathcal{L}_1(D_\psi, \mathcal{G}^k)$ as in equation 8
9:     **end for**
10:     Calculate task similarity $sim(t, k)$ using equation 5
11:     Initialize feature adaptor parameters $\phi_t$ and $\alpha_t$
12:     **if** $sim(t, k) > 0$ **then**
13:         Obtain $f_m^t$ of $\mathcal{G}^t$ using equation 7
14:         Initialise $\mathbf{U}^t$, $\mathbf{b}^t$ using weights from step 8
15:     **else**
16:         Obtain $f_m^t$ of $\mathcal{G}^t$ using equation 6
17:         Initialise $\mathbf{U}^t$, $\mathbf{b}^t$ using weights from step 4
18:     **end if**
19:     Optimize $\mathbf{U}^t$, $\mathbf{b}^t$, and $\phi^t$ using $\mathcal{L}_1(D_\psi, \mathcal{G}^t)$ as in equation 8
20: **end for**

---

## 4 Experiments and Results

### 4.1 Baselines and Metrics

We employ StyleGAN2 Karras et al. (2020a) as the base architecture for all experiments. StyleCL is compared to GAN Memory, CAM-GAN with task similarity learning, and MerGAN. We implemented all the baselines with the StyleGAN2 backbone to ensure a fair comparison. Evaluation metrics include the Fréchet Inception

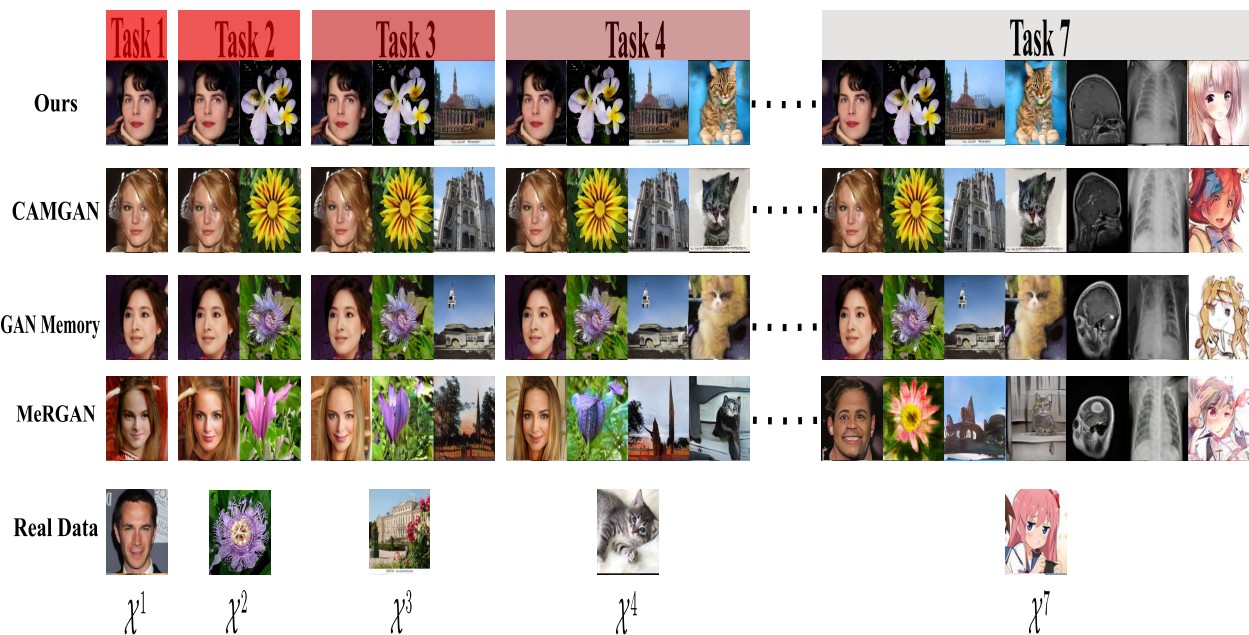

Figure 2: **Qualitative Evaluation:** Generated Images by StyleCL compared against various baselines across six distinct tasks, including the initial base task. Each row showcases results from a different generative method, while each column displays the progression of generated samples, illustrating the model's capacity to retain and refine image quality throughout the sequence of tasks. This visual representation highlights the competence of StyleCL in maintaining high-fidelity generation across diverse datasets.

Distance (FID) Heusel et al. (2017), Density, and Coverage Naeem et al. (2020), which are pivotal for assessing the quality and diversity of generated images. Additionally, we examine the computational and memory overhead using FLOPs and parameter count Dehghani et al. (2022), essential for evaluating the scalability of continual learning approaches. As pointed out in Dehghani et al. (2022), data from major cloud service providers indicate that 85-90% of ML workload is in inference processing, considering this we measure FLOPs during inference stage using fvcore package[1].

For comprehensive definitions of these metrics, please refer to Appendix B.2.

## 4.2 Results for perceptually distant tasks

Adhering to protocols established in CAM-GAN and GAN-Memory, our initial model training employs the CelebA-HQ Karras et al. (2018) dataset. This is followed by sequential training on six datasets that are significantly varied in visual content, thus 'perceptually distinct': Oxford 102 Flowers Nilsback & Zisserman (2008), LSUN Church Yu et al. (2015), LSUN Cats Yu et al. (2015), Brain MRI Cheng et al. (2016), Chest X-Ray Kermany et al. (2018), and Anime Faces [2]. Qualitative comparisons of generated samples (Fig. 2) demonstrate that StyleCL consistently produces superior image quality compared to the baselines. We provide additional qualitative results in Appendix B.2 of supplementary materials.

Quantitative results, detailed in Tab. 1, confirm that StyleCL surpasses the baseline methods on most fronts according to FID, Density, and Coverage metrics. Additionally, Tab. 2 illustrates StyleCL's reduction in parameters and the marginal increase in FLOPs, training and inference time for each of the methods. We provide an expanded version of scalability metrics in Appendix B.2 of supplementary materials. Despite lacking specific adaptation modules present in methods like CAM-GAN, StyleCL efficiently modulates the latent space, achieving high-quality generation with fewer parameters. While MerGAN maintains static

---

[1]https://github.com/facebookresearch/fvcore/blob/main/docs/flop_count.md
[2]https://github.com/jayleicn/animeGAN

parameters and FLOP counts, it sacrifices generation quality for previous tasks—an issue StyleCL circumvents with its no-forgetting approach.

| Methods | MerGAN | | | GAN Memory | | | CAM-GAN | | | StyleCL | | |
|---|---|---|---|---|---|---|---|---|---|---|---|---|
| Metrics / Dataset | FID ↓ | D ↑ | Cov ↑ | FID ↓ | D ↑ | Cov ↑ | FID | D ↑ | Cov ↑ | FID ↓ | D ↑ | Cov ↑ |
| **Flowers** | 45.14 | 0.6 | 0.49 | 23.97 | 0.73 | 0.71 | 23.38 | **0.89** | 0.71 | **18.48** | 0.67 | **0.77** |
| **LSUN Church** | 31.41 | 0.56 | 0.18 | 37.9 | 0.30 | 0.11 | 24.25 | 0.20 | 0.17 | **17.36** | 0.59 | **0.41** |
| **LSUN Cat** | 53.52 | 1.10 | 0.20 | 53.22 | 0.86 | 0.32 | 52.59 | 0.62 | 0.22 | **34.43** | 1.15 | **0.41** |
| **Brain MRI** | 78.80 | 0.16 | 0.29 | 45.78 | 0.32 | 0.55 | 31.26 | 0.18 | 0.77 | **29.42** | 0.38 | **0.82** |
| **Chest X-Ray** | 58.51 | 0.13 | 0.11 | 58.82 | 0.23 | 0.3 | **24.81** | 0.36 | 0.73 | 25.83 | 0.55 | 0.75 |
| **Anime** | 39.83 | 0.35 | 0.09 | 16.20 | **0.63** | 0.38 | 21.52 | 0.50 | 0.27 | **12.38** | 0.62 | **0.39** |

Table 1: Quantitative performance metrics of StyleCL compared with MerGAN, GAN Memory, and CAM-GAN across six datasets. Fréchet Inception Distance (FID), Density (D), and Coverage (Cov) are used as evaluation metrics, with the ideal direction of metric improvement indicated in parentheses. The table clearly demonstrates the superior performance of StyleCL in terms of lower FID scores and higher Density and Coverage across most datasets, signifying better image quality and greater diversity in generated samples.

| Algorithm | Params Increase per Task ↓ | FLOPs Increase (%) ↓ | Training Time (s) (per iteration) ↓ | Inference Time (ms) ↓ |
|---|---|---|---|---|
| GAN Memory | 4.21M | 15.7 | 230 | 7.32 |
| CAM-GAN | 1.52M | 23.32 | 180 | 4.74 |
| **StyleCL** | **1.08M** | **4.1** | **110** | **3.92** |

Table 2: Comparison of our approach (StyleCL) against the baselines GAN Memory and CAM-GAN regarding parameter increase per task, percentage increase in FLOPs, training and inference time.

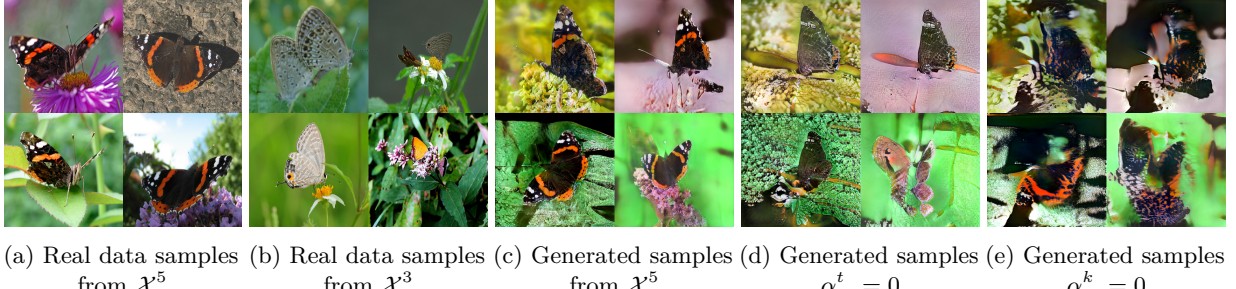

(a) Real data samples from $\mathcal{X}^5$  (b) Real data samples from $\mathcal{X}^3$  (c) Generated samples from $\mathcal{X}^5$  (d) Generated samples $\alpha_m^t = 0$  (e) Generated samples $\alpha_m^k = 0$

Figure 3: **Qualitative illustration of forward transfer in StyleCL**: Fig. 3a and Fig. 3c corresponds to real and generated samples from current task $\mathcal{X}^5$. StyleCL employs feature adaptors from previous tasks (samples of which are shown in Fig. 3b) to generate shared features across tasks (Fig. 3d). Meanwhile, it utilizes newly added feature adaptors exclusively for the unique features of the current tasks (Fig. 3e).

### 4.3 Results on perceptually similar tasks

In order to evaluate the forward transfer capability of StyleCL, we consider six varieties of butterflies from ImageNet to create a sequence of perceptually similar generation tasks, $\mathcal{X}^1$ to $\mathcal{X}^6$. We consider 2 scenarios : (a) StyleCL that enables forward transfer by considering the generator of the most similar previous task, and (b) StyleCL with parameter sharing only with the base task $\mathcal{G}^1$ (without forward transfer). Tab. 3 summarizes the results for both scenarios. Additional comparisons for more diverse datasets are given in section B.4 of the supplementary material. We observe improved performance on most datasets for scenario (a) compared to scenario (b), confirming the benefit of forward transfer, inherent in our method. Also, the amount of knowledge that could be reused varies (positive or negative forward transfer) which leads to varying degrees

of improvement. As observed from Tab. 3 in case of $\mathcal{X}^3$, $sim(t,k) < 0$ indicates potential negative forward transfer and hence when the model is forced to reuse the most similar task, it results in a performance drop. This empirically validates our characterization of the nature of forward transfer by using $sim(t,k)$. In such cases, we prevent negative forward transfer by avoiding parameter reuse from the most similar task.

To qualitatively evaluate the forward transfer capability of our approach, StyleCL, we train it on dataset $\mathcal{X}^5$ shown in Fig. 3a using the generator of the most similar previous task, $\mathcal{X}^3$ whose samples are shown in Fig. 3b. The generated samples are illustrated in Fig. 3c. To analyze the individual contribution of current and previous feature adaptors in StyleCL, we separately disable their individual contribution by setting $\alpha_m^t$ and $\alpha_m^k$ to 0 in equation 7. The corresponding generated samples are illustrated in Fig. 3d and Fig. 3e. Our results show that $\phi_m^3$ is reused to capture shared characteristics of $\mathcal{X}^5$ and $\mathcal{X}^3$, such as shape and background (as seen in Fig. 3d), whereas newly introduced feature adaptors $\phi_m^5$ capture features unique to $\mathcal{X}^5$, such as the orange color of the wings (as seen in Fig. 3e). These findings confirm that StyleCL enables forward transfer by reusing knowledge from previous tasks.

|                          | $\mathcal{X}^2$ | $\mathcal{X}^3$ | $\mathcal{X}^4$ | $\mathcal{X}^5$ | $\mathcal{X}^6$ |
|--------------------------|-------|---------|-------|---------|---------|
| StyleCL without transfer | 28.96 | **35.68** | 20.90 | 27.51 | 31.84 |
| StyleCL with transfer    | **21.86** | 37.38 | **18.87** | **23.18** | **31.00** |
| $sim(t,k)$               | 0.59  | -0.02   | 0.21  | 0.35    | 0.37    |

Table 3: Comparison of StyleCL with and without forward transfer on a stream of perceptually similar datasets denoted as $\mathcal{X}^i$, $2 \leq i \leq 6$. Results are reported in terms of FID, where lower values indicate better performance.

## 5 Analysis and Ablations

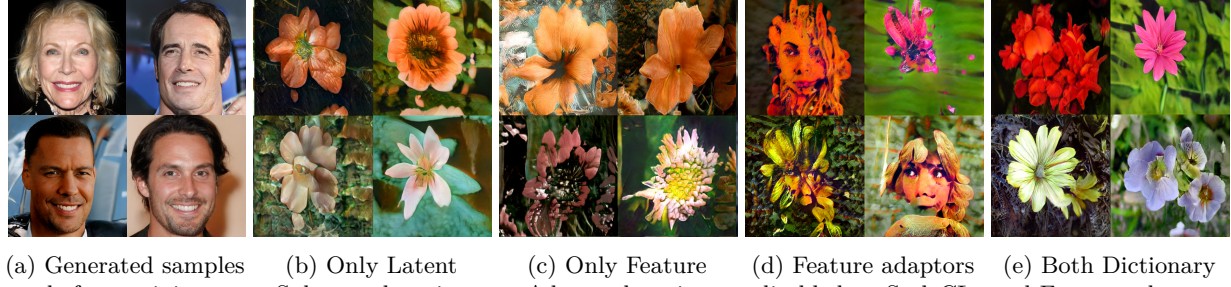

| (a) Generated samples before training | (b) Only Latent Subspace learning | (c) Only Feature Adaptor learning | (d) Feature adaptors disabled on StyleCL | (e) Both Dictionary and Feature adaptors. |

Figure 4: **Ablation Study on Feature Adaptors and Latent Dictionary**. Fig. 4a shows initial generations without training. Learning only the latent subspace (Fig. 4b) or the feature adaptors (Fig. 4c) captures partial data characteristics. The combined approach (Fig. 4e) yields high-quality, artifact-free samples. Disabling feature adaptors during StyleCL inference (Fig. 4d) still preserves certain data traits, such as color and background, demonstrating the effectiveness of the latent dictionary in capturing the essence of the data.

### 5.1 Analysis of Task-Specific Latent Subspaces

The learned task-specific latent dictionary vectors, denoted as $\mathbf{U}^t$, characterizes the task-specfic latent subspaces and is pivotal for choosing the most similar task and averting negative forward transfer. We employ t-SNE visualizations to empirically validate the capacity of these latent subspaces to encapsulate task semantics. Utilizing equation 1 and equation 2, we sample latent vectors $\mathbf{w}^t$ from the latent subspaces for a curated set of tasks, aiming to show that vectors from related tasks congregate, whereas those from unrelated tasks diverge.

Specifically, we select two semantically similar Butterfly datasets (refer to Sec. 4.3) and a contrasting Brain-MRI task (see Sec. 4.2) for visualization. Fig. 5 depicts the t-SNE plots, where one can observe the clustering of latent vectors within the space; vectors from related tasks (Butterfly 1 in blue, Butterfly 2 in orange) are proximal, whilst vectors from the unrelated task (Brain MRI in green) are isolated. This spatial distribution substantiates our hypothesis that the learned latent subspaces indeed reflect task semantic similarities and supports their utility in identifying similar tasks and discerning the nature of forward transfer.

| Dataset | Butterfly 1 | Butterfly 2 | Brain MRI |
|---|---|---|---|
| Butterfly 1 | 1 | 0.6 | 0.14 |
| Butterfly 2 | 0.6 | 1 | 0.15 |
| Brain MRI | 0.14 | 0.15 | 1 |

Table 4: Average cosine similarity of dictionary vectors for similar and dissimilar tasks

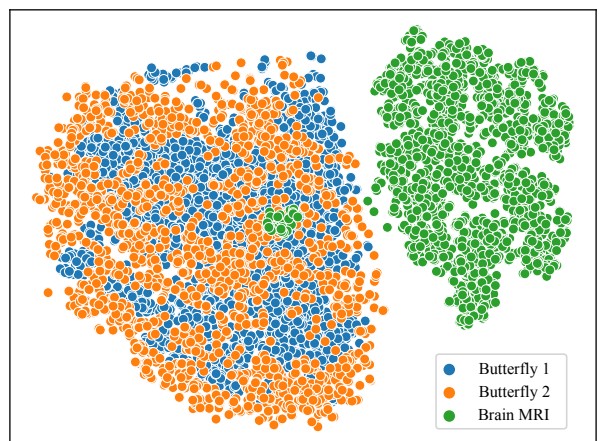

Figure 5: **t-SNE visualization of latent vectors:** Demonstrating the distinction and proximity of latent vectors within learned subspaces. Latent vectors from semantically related Butterfly datasets are depicted in blue and orange, showcasing their clustering, while vectors from the dissimilar Brain-MRI task are in green, indicating significant separation. This illustrates the latent spaces' capability to differentiate and group tasks based on semantic relatedness.

Furthermore, the parameter $K$, representing the number of dictionary vectors per style block, is a critical design variable. An in-depth analysis of $K$'s influence on the generative quality and the selection rationale behind it is presented in Section C.2 of the supplementary materials.

## 5.2   Ablation Studies on Latent Dictionary and Feature Adaptors

To dissect the individual contributions of our method's components, we conduct ablation studies focusing on the learned latent dictionary and feature adaptors. StyleCL is trained using each module independently, and the outcomes are qualitatively assessed in Fig. 4. The isolated effects of dictionary learning (Fig. 4b) and feature adaptor learning (Fig. 4c) yield lower quality generations compared to utilizing both modules in tandem (Fig. 4e).

Furthermore, we explore the influence of dictionary learning on StyleCL during the inference stage. By deactivating the current feature adaptor's output and retaining the rest of the model structure, we observe that the generated samples (Fig. 4d) retain certain attributes, such as color and background, akin to the complete StyleCL-generated samples (Fig. 4e). This phenomenon substantiates our hypothesis that StyleCL can effectively approximate the target data distribution with fewer parameters, primarily attributed to the integration of dictionary learning. A qualitative comparison is presented in Appendix B.3, complemented by further qualitative evaluations across additional datasets.

### 5.3 Impact of Base task on StyleCL

The generator $\mathcal{G}^1$, pivotal for all subsequent tasks, is initially trained on the $\mathcal{X}^1$ dataset. To understand how different base task affect the efficacy of StyleCL, we conduct experiments with the base task being Brain MRI and ImageNet datasets. We assess the performance of StyleCL on a data sequence comprising CelebA-HQ, Flowers, LSUN Church, and Chest X-Ray datasets under these conditions. The results, detailed in Tab. 5, highlight a notable improvement in performance when the base task is a heterogeneous dataset like ImageNet-1K, as opposed to starting with a more specialized dataset like Brain MRI. This indicates that StyleCL significantly benefits from initial training on a dataset with broader diversity. We believe that this could be attributed to diverse features captured by $\mathcal{G}^1$ which is subsequently useful for other tasks.

| Base Task | CelebA-HQ | Flowers | LSUN Church | Chest X-Ray |
|---|---|---|---|---|
| Brain-MRI | 22.82 | 31.98 | 55.45 | 29.93 |
| ImageNet | **15.86** | **14.25** | **11.71** | **23.54** |

Table 5: Performance of StyleCL with different base task $\mathcal{X}^1$, measured by Fréchet Inception Distance (FID) across four datasets. Lower FID values indicate better image quality and higher similarity to the target dataset. ImageNet initialization outperforms Brain-MRI across all datasets, affirming the advantage of diverse initial training.

## 6 Conclusions and Future Work

We introduce StyleCL, a novel generative continual learning approach utilizing the latent space of StyleGAN. This strategy employs task-specific latent subspaces and feature adaptors, circumventing the need for direct feature or weight modifications and thus minimizing computational and memory demands while boosting performance. Uniquely, it also prevents negative forward transfer, a previously unexplored aspect in this field. Future work aims to extend the applicability of StyleCL to various architectures and generative models and improve continual learning through shared latent dictionaries. We will also focus on refining StyleCL for use in scenarios without clear task identification, broadening its practical real-world application. Our work opens new avenues for the efficient and effective training of generative models, with anticipated enhancements solidifying the role of continual learning in generative model evolution.

### Acknowledgments

Dr. Prathosh A.P would like to acknowledge the support provided by the Indian Institute of Science, for setting up the compute infrastructure. He also would like to thank TCS Research for collaborations.

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
