# OpenReview forum: "Lifelong Learning in StyleGAN through Latent Subspaces"
_TMLR — Accepted by TMLR_

### Review · Reviewer_nhoR · 2024-05-20

**Summary Of Contributions:**

This submission proposes a continual learning paradigm for StyleGAN over a variety of tasks with versatile visual content, called StyleCL. Specifically, the input images are first used to fit a base generator $\mathcal{G}^1$. The latent subspace is then initialized and learned on $\mathcal{G}^1$. Next, for all subsequent tasks, the authors propose to identify the most similar task to the current one and then calculate the cosine similarity between the dictionary vectors and their projections onto the latent subspace to prevent negative forward transfer. Additionally, the authors introduce a learnable feature adaptor in each style block to compute the feature activations. The authors then test StyleCL on a sequence of visual tasks to showcase the adaptability of StyleCL to different visual contents across tasks. Ablation studies are also conducted to investigate the task semantics in the latent dictionary vectors, the necessity of feature adaptors, and the role of base generator $\mathcal{G}^1$.

**Audience:**

Yes

**Broader Impact Concerns:**

I do not see immediate ethical concerns about the work presented, though I think the authors should briefly discuss the potential limitations of the proposed method and the future direction of research.

**Claims And Evidence:**

No

**Requested Changes:**

•	For better illustration, it would be better to visualize the discriminator $D_{\psi}$ in Figure 1.

•	More clarifications on lines 6-8 in Algorithm 1 (see the 2nd point under weaknesses) would be very helpful.

•	To empirically demonstrate the training efficiency of StyleCL (see the 3rd point under weaknesses), it would be helpful to compare the training and inference times with those of other GAN-based benchmarks in the experiments.

•	In Figure 5, why do a small portion of t-SNE vectors of Brain MRI lie within the cluster of vectors of butterflies? Does this mean $\textbf{U}^t$ still detects some similarities between these two types of visual tasks?

**Strengths And Weaknesses:**

Strengths:

•	Continual learning for generative models is important and under-explored. The topic is potentially of interest to the TMLR community.

•	The motivation of subspace learning based on task similarity is reasonable based on the idea of knowledge transfer in continual learning.

•	StyleCL shows good performance on a sequence of perceptually distant tasks compared to other GAN-based benchmarks based on the provided visualizations and quantitative results, which serves as convincing evidence for the applicability of the proposed method.

•	The authors conduct a variety of ablation studies to help explain the rationale of the key components of StyleCL, which is commendable.

•	The paper is generally easy to follow and well organized.

Weaknesses:

•	The rationale behind the recursive expression of $S_m^k (f_m^{t-1})$ in Equation 7 is not clearly explained. To be specific, how can we guarantee that activations from similar tasks of $k$ also result in a positive forward transfer?

•	Lines 6-8 in Algorithm 1 are somewhat confusing. After learning $\textbf{U}^t$ and $\textbf{b}^t$ using $\mathcal{L}_1 (D\_{\psi}, \mathcal{G}^1)$, why cannot we determine $k$ based on the bias vector $\textbf{b}^t$ and then immediately calculate $sim(t,k)$ based on the dictionary $\textbf{U}^t$ and its projections? That is, why cannot we combine the steps of identifying the most similar task and preventing negative forward transfer?

•	Despite the reduction in parameters, the training process of StyleCL does not seem to be very efficient. Specifically, in Algorithm 1, $\textbf{U}^t$ and $\textbf{b}^t$ are optimized using the GAN loss 3 times for each task $t = 2, …, T$.

---

> ### Author Response · Authors · 2024-06-02
> **Response to Reviewer nhoR**
>
> We thank the reviewer (R1: nhoR) for the valuable feedback. We address all the concerns in this rebuttal and will revise the paper accordingly.
>
> **Q1: How can we guarantee that activations from similar tasks of k also result in a positive forward transfer?**
>
> $sim(t,k)$ ensures that the combined contributions of all similar tasks of $k$ lead to a positive forward transfer to task $t$, thereby preventing any negative forward transfer, even though activations from individual tasks may not always result in a positive forward transfer.
>
> **Q2: Why can't we combine the steps of identifying the most similar task and preventing negative forward transfer?**
>
> To identify the most similar tasks, we ensure that all task vectors lie in the shared latent space of the generator $\\mathcal{G}^1\$, while $sim(t,k)$ is computed using the latent vectors of $\\mathcal{G}^k\$. Computing $sim(t,k)$ using latent vectors from the similar tasks identification step does not capture the nature of forward transfers due to other similar tasks of task $k$. Hence, the two steps cannot be combined.
>
> **Q3: Compare the training and inference times with those of other GAN-based benchmarks in the experiments**
>
> Comparison of training and inference times:
>
> |                | Training time (s) (per iteration) | Inference time (ms) |
> |----------------|-----------------------------------|---------------------|
> | GAN Memory     | 230                               | 7.32                |
> | CAM-GAN        | 180                               | 4.74                |
> | StyleCL        | 110                               | 3.92                |
>
> **Q4: Why do a small portion of t-SNE vectors of Brain MRI lie within the cluster of vectors of butterflies?**
>
> We were unable to definitively determine the reason for this, as visualizations of these samples did not demonstrate any similarity. We believe these are spurious clusters and do not correspond to actual similarities.

---

### Review · Reviewer_oQcf · 2024-06-05

**Summary Of Contributions:**

This paper tackles the problem of lifelong or continual learning in the context of GANs. By lifelong learning, it refers to learning over a number of datasets and tasks, which might differ in their distributional characteristics. In this connection, catastrophic forgetting is also pertinent, as the network might forget what it has learnt when new tasks or datasets arrive in the sequence.

The method consists of building upon StyleGAN2 to construct a dictionary $U_m^t$ for each task $t$, consisting of latent subspace vectors $u_{mk}$ for each of the StyleGAN codeblocks $m$. To create the latent style vector to pass into StyleGAN2, they sample from each of the latent subspace vectors, and add a bias term to it. After learning on the most recent task, it is compared with the previous tasks by using a similarity metric that checks the difference between the bias vectors. This gives us the 'most similar' generator. This most similar generator is then evaluated for fitness to see if it contributes positively or negatively in the task at hand. If it is positive, the generator in question is picked and its subspace parameters are updated. If not, they go with the 'base' generator. To further improve results, the model also has an adaptor component in weight space.

Evaluations are presented over a number of datasets, ablations and baseline methods. Mainly, they show that their method shows improved results in FID, density and coverage. Ablations on model components are also shown.

**Audience:**

Yes

**Claims And Evidence:**

Yes

**Requested Changes:**

Can the authors expand on the reasoning for picking the 'bias' term to get similarity metric? On similar lines, some more explanation or evidence for negative subspaces and feature adaptor would help. This could be, for instance, from a carefully designed experiment that shows the effect of these fittings.

**Strengths And Weaknesses:**

Strengths
-------------
+ Paper is well written and all components are explained in a comprehensible way
+ Results bear out the claims and code is provided to validate.
+ The idea of 'mixing' latent subspaces and obtaining similar task scores is intuitively appealing.

Weaknesses
-----------------
- While I can see that the results are convincing, there are some aspects that concern me. The theoretical reasoning for comparing biases to measure similarity is not clear. Likewise, using the Gram-Schmidt procedure is sensible but looks arbitrary - here we are omitting the bias vectors as well.
- The impact of 'negative' subspaces could be quantified.
- I am not very sure how the feature adaptor helps. This is again something that makes intuitive sense, but perhaps needs more insight.
- The aspect of catastrophic forgetting is mentioned, but no evidence is given in the paper to show that the fittings address this issue.

---

> ### Author Response · Authors · 2024-06-17
> **Response to Reviewer oQcf**
>
> We appreciate Reviewer R2 (oQcF) for the insightful comments and valuable feedback. We have addressed all concerns as follows:
>
> **Evidence of Effectiveness of Feature Adaptor:** We direct the reviewer to Fig. 2 (right) in the supplementary material, which quantitatively evaluates the effect of the feature adaptor on StyleCL's performance. As observed in Fig. 2, the generation quality, measured by FID, is relatively poorer without the feature adaptor (indicated in red). Upon incorporating the feature adaptor, the generation quality improves, resulting in a lower FID (indicated in green). This demonstrates the significant impact of the feature adaptor on StyleCL's performance.
>
> **Overcoming Catastrophic Forgetting:** To demonstrate that the proposed method overcomes catastrophic forgetting, we conducted an ablation study comparing the performance of StyleCL to naive fine-tuning. The quantitative comparison is provided below.
>
> |                | $\mathcal{X}^2$ | $\mathcal{X}^3$ | $\mathcal{X}^4$ | $\mathcal{X}^5$ |
> |----------------|-----------------|-----------------|-----------------|-----------------|
> | StyleCL        | 28.96           | 35.68           | 20.90           | 27.51           |
> | Naive Fine-tuning | 216         | 220             | 153             | 202             |
>
> *Table 1: Comparison of StyleCL with naive fine-tuning to demonstrate that StyleCL overcomes catastrophic forgetting. We train two models, one via naive fine-tuning and the other using StyleCL, on datasets $\mathcal{X}^i$, $2 \leq i \leq 6$, and tabulate the generation quality on previous tasks after training till $\mathcal{X}^6$ continually. Results are reported in terms of FID, where lower values indicate better performance.*
>
> As observed in Table 1, with naive fine-tuning, the model forgets all previous tasks, leading to high FID scores. In comparison, StyleCL retains performance on previous tasks, thereby showing its capability to overcome catastrophic forgetting.
>
> **Evidence for Negative Subspaces:** To illustrate the effect of negative subspaces, we refer the reviewer to Table 3, column 2, where $sim(t,k)$ is slightly negative. In this scenario, forcing StyleCL to use negative subspace results in the FID worsening from 35.68 to 37.38, qualitatively demonstrating that negative subspaces harm generation quality.
>
> **Reasoning for Selecting the 'Bias' Term to Obtain Similarity Metrics:**
>
> To theoretically motivate the comparison of biases for measuring similarity, consider a simplistic scenario where the dictionary consists of a single vector and a bias vector. In this case, the latent vectors lie in an affine subspace: $b + \alpha u_1$. Similarly, we could have a latent space represented as $b' + \alpha' u_1'$ for another dataset.
>
> To find the most similar subspace for a given task, we start by computing the Euclidean distance between the bias vectors $b$ and $b'$. This distance measures how close the subspaces are in the feature space. After this, we can further measure the alignment between the subspaces by evaluating the similarity metric $sim(t,k)$ which compares the orientation and overlap of the subspaces defined by the respective dictionaries.
>
> By focusing on the bias terms, we effectively capture a key aspect of the subspace's positioning within the overall latent space, allowing for a meaningful comparison between different datasets.

---

> > ### Comment · Reviewer_oQcf · 2024-10-30
> > **On clarifications**
> >
> > Thanks for addressing all my questions.

---

### Review · Reviewer_XjRr · 2024-10-06

**Summary Of Contributions:**

The paper proposes a method for lifelong learning of styleGAN, which can reuse the trained parameters on previous tasks.

**Audience:**

Yes

**Claims And Evidence:**

Yes

**Requested Changes:**

See weaknesses.

**Strengths And Weaknesses:**

### Strengths:

* The lifelong learning problem and the proposed solution are practical and interesting.
* The algorithm and mathematical formulations are clear.
* The presented experiment results demonstrate that the proposed method is effective, especially the quantitative part.


### Weaknesses:

* As for Fig.2 for qualitative evaluation, it seems that no obvious difference presented between four methods.
* The lifelong learning ability of the proposed method needs to be further clarified or evaluated. The proposed method targets to solve the lifelong learning problem, so that the results relative to $t$ is a very important aspect in quantitative evaluation. In the paper, Tab. 1 demonstrates that the proposed method achieves good performance throughout the process of increasing $t$. However, it is not clear how much the starting point(when t=0) contributes. These baselines perform poorly from the initial task to the final task compared to the proposed method, it is difficult to determine whether this is due to the poor performance of the generative model itself or the poor lifelong learning ability.


### Minor problems:

1).Define StyleCL before use it in abstract.
2).  Problem setting - > 3.1 Problem Setting

---

> ### Author Response · Authors · 2024-10-15
> **Response to Reviewer XjRr**
>
> We thank the reviewer for the valuable time and suggestions. We address some of the concerns raised by the reviewer below.
>
> ### W1. Imperceptible changes in qualitative results
>
> **A1.** We thank the reviewer for pointing this out. The imperceptible difference is mainly due to the samples being randomly selected, as well as the relatively small number of samples shown in **Figure 2**. We have updated **Figure 2** by sampling more representative images to reflect the average image quality more accurately.
>
> ---
>
> ### W2. Difficulty in determining whether this is due to the poor performance of the generative model or the poor lifelong learning ability
>
> **A2.** We appreciate the reviewer’s query regarding whether the improved performance of StyleCL is due to a better generative model or its improved lifelong learning ability compared to the baselines. For all baselines and StyleCL, we use the same generative model, StyleGAN2, as the backbone. Therefore, any improvement in StyleCL’s performance cannot be attributed to a better generative model but rather to its enhanced lifelong learning ability.
>
> ---
>
> ### Minor problems
>
> Thank you for pointing these out; we have fixed them in the latest revision.

---

### Review · Reviewer_VJrr · 2024-10-08

**Summary Of Contributions:**

In this paper, the authors propose to leverage the versatility of StyleGAN to enable long-term learning of StyleGAN without forgetting. Their method SOTA approaches while employing fewer parameters and FLOPS. Not only that, but their framework prevents negative forward transfer. According to what they said, StyleCL is the only method capable of averting negative forward transfer.

**Audience:**

Yes

**Claims And Evidence:**

No

**Requested Changes:**

Please check the weaknesses section above.

Minor typo (maybe):

Section 3.1: subsequent datasets dataset $X^t$?
Section 3.2:  StyleGAN2 Karras et al. (2020a) --> StyleGAN2 (Karras et al., 2020a)?

**Strengths And Weaknesses:**

Strength:
1. Their paper is relatively clearly written, although I think there are some points that I wish to be revised. At least, I like that they included pseudocode and the overall figure to help understand readers.
2. They emphasized their novelty throughout Section 3, which helped me easier to understand their contributions (such as finding out task similarity using cosine measure)


Weakness:

1. Though it seems like the authors tried their best to explain their models using images, I think it would be better if there's more detailed explanation for each subfigure in Figure 1. It would be a great idea to split this one figure into three, and if possible make some 'flow arrow' that directs 'this $G^t$ in Figure 1(a) (the largest concept) is constructed as Figure 1(b) ...' with a bit more detailed explanation. For example, what is that $m \times d \times K$ cube? What is the dimensionality of $f_m^t$ and the style block $S_m^t$? (For the beginners of this field it might be tough to understand what is going on. Maybe adding StyleGAN basics in the Appendix, for completeness?)

2. What happens if the first dataset $\mathcal{X}_1$ follows a skewed distribution? It seems like if the algorithm fails to find another similar task, then it relies on the first model for initialization, but if it has some.... poor quality, then I think it might affect the overall result.

3. To be honest, now I have difficulty in understanding your Section 3.4.
3-1) Why do we need to talk about style block $S_m^t$ and activation $f_m^t$? Why $U$ is not enough? The first paragraph of 3.4 is not enough for me to understand what is going on. Is it empirical things?
3-2) It literally copies the parameters from $k$-th trained model in Eq. (7) without considering similarity (I think the new model should copy less if the similarity is low).

---

> ### Author Response · Authors · 2024-10-15
> **Response to Reviewer VJrr**
>
> We thank the reviewer for the comments and questions. Below we address the main concerns raised.
>
> ### W1. Further clarification on the block diagram and the addition of StyleGAN2 basics in the appendix
>
> **A1.** As suggested by the reviewers, we have further modified the block diagram to incorporate the recommended changes. These changes include more explicit segregation of components, a detailed explanation for each component, and the addition of directional arrows to improve the flow. Also, we have added a brief discussion on StyleGAN and architectural details in the appendix.
>
> ---
>
> ### W2. What happens if the first dataset follows a skewed distribution?
>
> **A2.** As the reviewer rightly pointed out, the performance of StyleCL in scenarios where the model is unable to find a similar task largely depends on the base model's performance. We have quantified this impact in **Section 5.3**. Specifically, StyleCL's performance diminishes when the distribution of $X_1$ is skewed (e.g. when $X_1$ is derived from a domain such as medical data). This is due to the limited number of transferable features across domains in such cases.
>
> A potential solution to mitigate this issue is to pretrain the generative model on a large, balanced dataset such as **ImageNet** before introducing the continual data stream. As demonstrated in **Table 5, Row 2**, using such a pre-trained model as the base task significantly improves performance.
>
> ---
>
> ### W3. Requirement of $S_m^t $ and selective copying
>
> **A3.1.** $U$ modulates the existing knowledge in the generator to facilitate generation from the current task. However, incoming data streams may be from vastly different domains. Modulating already learned features from previous tasks may not be sufficient for good generation from the current task's dataset.
>
> To accommodate this, we utilize the style block $S_m^t$ and its corresponding activations. This is further validated empirically, as shown in **Figure 1** of the appendix. We observe that merely using $U$ results in a relatively high FID, while using $S_m^t$ reduces the FID, thus enhancing generation quality. We will add these references to the beginning of **Section 3.4** to avoid confusion.
>
> **A3.2.** We appreciate the reviewer's suggestion of copying the weights proportional to similarity, which could reduce the number of parameters. This approach could be seen as a continuous variant of our current binary approach (which either copies the previous weights or discards them).
>
> However, we did not incorporate this into our framework primarily due to the additional computational cost and complexity associated with similarity attribution in the weight space (as opposed to in the latent space, as in the case of ${sim}(t,k)$ to estimate which neurons from the previous task positively or negatively affect the current task.
>
> ---
> Minor changes: We have updated the main text to rectify these minor issues.

---

### Decision · Action_Editor_CVoe · 2024-11-11

**Recommendation:** Accept with minor revision

**Comment:**

In a private discussion, one of the reviewers expressed the following (I'm paraphrasing):

The reviewer had difficulty understanding the paper’s approach, even after the revision, particularly the theoretical need for style blocks and activation functions. The reviewer mentioned that the paper tried to explain these aspects based on empirical observations rather than theory. They also mentioned that the similarity measure is used indirectly through the model selection, and so couldn't understand precisely how it affected the results. Consequently, they were leaning toward rejection, but said that they would be ok if this paper got accepted.

**Audience:**

Yes, definitely. It is likely to be of interest to both the continual learning community and the GAN community.

**Claims And Evidence:**

This paper presents a method for continual learning in GANs, specifically using the latent space in StyleGAN to incrementally learn without catastrophic forgetting. The reviewers were generally positive, noting that the paper addresses a significant and under-explored problem in lifelong learning for generative models. Overall, the reviewers found the paper to be innovative and practical, supported by strong empirical results that showed competitive performance against the relevant baselines. In their revisions, the authors addressed most of the concerns reviewers raised in their initial reviews.

While most reviewers found the paper clear and well-organized, one reviewer expressed concerns regarding the theoretical motivation for particular components, particularly the style blocks and activation functions. Additionally, other reviewers suggested that the paper would benefit from deeper theoretical explanations, and there were several other related more-mild concerns, such as the explanation of Equation 7, that it is unclear whether they were sufficiently addressed in the authors’ earlier revision. Concerns about computational efficiency were raised as well, given the iterative GAN optimization for each task.

Overall, the paper’s strengths in addressing an important problem, along with its empirical robustness, support its acceptance. My recommendation is therefore to Accept with Minor revisions, where the authors should focus on clarifying theoretical motivations and computational considerations as raised in the reviews.